# Prevention of Peripherally Inserted Central Catheter (PICC)-Associated Vein Thrombosis in Cancer: A Narrative Review

**DOI:** 10.3390/biomedicines13040786

**Published:** 2025-03-24

**Authors:** Agnese Maria Fioretti, Pietro Scicchitano, Daniele La Forgia, Raffaele De Luca, Elena Campello, Carlo Gabriele Tocchetti, Marcello Di Nisio, Stefano Oliva

**Affiliations:** 1Cardio-Oncology Unit, IRCCS Istituto Tumori, “Giovanni Paolo II”, 70124 Bari, Italy; a.fioretti@oncologico.bari.it (A.M.F.); s.oliva@oncologico.bari.it (S.O.); 2Cardiology-Intensive Care Unit, Ospedale della Murgia “Fabio Perinei”, Altamura, 70022 Bari, Italy; 3Breast Radiology Department, IRCCS Istituto Tumori “Giovanni Paolo II”, 70124 Bari, Italy; d.laforgia@oncologico.bari.it; 4Department of Surgical Oncology, IRCCS Istituto Tumori “Giovanni Paolo II”, 70124 Bari, Italy; dr.raffaele.deluca@gmail.com; 5Internal Medicine, Hemorrhagic and Thrombotic Diseases Unit, Department of Medicine (DIMED), Padova University Hospital, 35121 Padova, Italy; elena.campello@unipd.it; 6Department of Translational Medical Sciences (DISMET), “Federico II” University of Naples, 80131 Napoli, Italy; cgtocchetti@gmail.com; 7Center for Basic and Clinical Immunology Research (CISI), “Federico II” University of Naples, 80131 Napoli, Italy; 8Interdepartmental Center of Clinical and Translational Sciences (CIRCET), “Federico II” University of Naples, 80131 Napoli, Italy; 9Interdepartmental Hypertension Research Center (CIRIAPA), “Federico II” University of Naples, 80131 Napoli, Italy; 10Department of Medicine and Ageing Sciences, “G. d’Annunzio” University of Chieti-Pescara, 66100 Chieti, Italy; mdinisio@unich.it

**Keywords:** cancer-associated thrombosis, central vein catheter, port, anticoagulation, prophylaxis

## Abstract

Venous thromboembolism (VTE) is considered the most common and potentially life-threatening cardiovascular complication in cancer and the second leading cause of death after cancer progression itself. In recent years, the steadily increasing rate of cancer-associated thrombosis (CAT) seems mainly related to amelioration in imaging techniques and the placements of central venous catheters (CVCs). The pivotal role of CVCs in the switch from hospital to home care is offset by its high thrombotic burden. The peripherally inserted central catheter (PICC) offers advantages (convenience, fast access, and cost-effectiveness) in comparison to centrally inserted devices (PORT), but increased thrombotic risk is reported. The aim of this narrative review was to offer a comprehensive overview of the existing literature about PICC-related thrombosis (PICC-VTE) by analyzing the current knowledge and related gaps. We further discussed advancements in insertion techniques, underscored the role of the novel PICC-PORT lines, and provided a “head-to-head” comparison among major guidelines on primary thromboprophylaxis.

## 1. Introduction

VTE is strongly associated to mortality, morbidity, healthcare expenditure, and a reduced quality of life in patients with cancer [1]. The high risk of VTE in these patients may be ascribed to hypercoagulable milieu and concomitant prothrombotic risk factors, including central venous catheters (CVCs) [2], which move hemostatic balance towards coagulation cascade activation [3]. The increase in CVC use for the long-term administration of chemotherapy and blood sampling is primarily driven by the desire to avoid painful venipunctures but is counterbalanced by an augmented risk for CVC-related VTE (CVC-VTE) with a higher rate in patients with PICC as compared to those with PORT [4,5,6]. Study heterogeneity in terms of population, design, detection techniques, and anticancer agents mirrors the vast variation in CVC-VTE incidence rate (1.5–71.9%) [7]. Most of the CVC-VTEs might occur as asymptomatic, while the rate of symptomatic is about 4 to 8% [8]. Indeed, advantages of PICC, including the possibility of ambulatory care, bedside insertion, nurse-led teams [9], prompt insertion/removal, and cost-effectiveness, still persist over PORT, despite the higher VTE risk [10,11,12,13,14,15]. The CAVA trial found that PICCs were associated with a five-fold higher likelihood of thrombosis as compared to PORT, which was explained by a smaller caliber arm over a longer length [16,17]. PICC-VTE appears intertwined with infections, including the setting of hospitalized patients, both in solid cancer [18] and hematological patients [19], fostered by their immunocompromised state, with the consequence of a mandatory catheter removal. Despite thromboprophylaxis feasibility [20], the ideal preventive approach remains highly debated [21]. There is lack of high-quality evidence about long-term management, and complications [22]. The aims of this narrative review were to evaluate current knowledge on PICC use, compare guidelines, and focus on critical preventive aspects: risk factor assessment, risk stratification models, primary thromboprophylaxis, and the use of the novel PICC-PORT lines.

## 2. Methods

We searched PUBMED (https://pubmed.ncbi.nlm.nih.gov/?otool=iitbisamlib, accessed on 11 November 2024), using the following key words: “vein OR venous”, “thrombosis OR clot OR thrombus”, “peripherally inserted central catheter OR PICC“, and “cancer OR tumor OR malignancy OR neoplasm”. Two reviewers screened all studies obtained by the search and extracted data from those of interest for the review. Articles written in languages other than English and those on children were excluded.

## 3. Risk Factors for the Development of VTE in Cancer Patients with PICC

Risk factors contributing to PICC-VTE are not completely defined and their further exploration is of utmost importance. Studies appraising the development of VTE in cancer patients with PICC were mostly retrospective, and had a small sample size, different outcome definition, and heterogeneous follow-up duration [23]. A large metanalysis including 5636 cancer patients found a higher rate of VTE in PICC as compared to PORT recipients (Odds Ratio OR: 0.43, Confidence Interval 95% CI 0.23–0.80); the risk further increased by previous VTE history, subclavian vein access, and improper tip position [24]. Lee and collaborators [25] found that 4.3% of their 444 cancer patients with PICC developed symptomatic CVC-VTE. Three significant risk factors for CVC-VTE were identified: ovarian cancer, ≥1 attempt at insertion, and previous CVC. In a large metanalysis including 29,503 patients, deep vein thrombosis (DVT) was more common in patients with PICC as compared to other CVCs (OR: 2.55, 95% CI 1.54–4.23, *p* < 0.0001), especially in those with cancer (6.67%, 95% CI 4.69–8.64) [26]. A retrospective study showed that a history of chemotherapy, manipulation nearby the catheter, and diabetes were the key risk factors for PICC-related thrombosis in cancer [27]. Indeed, obesity and lower daily activity from patients’ concern of PICC dislodgment were also independent risk factors for VTE-related thrombosis (OR: 3.466, *p* = 0.014; OR: 9.583, *p* = 0.000, respectively) [28]. A recent systematic review confirmed the role of a higher BMI in PICC-VTE occurrence: overweight/obese patients (BMI ≥ 25 Kg/m^2^) had a two-fold risk of PICC-VTE as compared to those with BMI < 25 Kg/m^2^ (28% vs. 13%, pooled relative risk [RR]: 2.06, 95% CI 1.21–3.49, *p* < 0.001) [29]. In a prospective study evaluating the risk of VTE in patients with CVC, PICC was associated with the highest risk of thrombosis (HR: 22.2, 95% CI 2.9–170.6). Age < 50 years, and previous VTE were predictive for thrombosis, whereas the jugular vein was the safest puncture site, possibly due to the larger vein lumen diameter [30]. Red cell distribution width (RCDW) also emerged as a potential risk factor as higher RCDW values significantly correlate with thrombosis. The same study also found an association between PICC-related thrombosis and smoking. Tobacco use could promote an increase in plasma concentration of fibrinogen, coagulation factors II, V, VIII, X, and XIII, tissue factor, and homocysteine, stimulating platelets activation/aggregation and fibrinolysis impairment [31]. These findings were corroborated by a recent metanalysis of 8635 patients: BMI ≥ 25 Kg/m^2^, D-dimer > 500 ng/mL, increased fibrinogen, elevated platelet count, and catheter malposition were independently related to PICC-VTE [32]. Beyond age, BMI, gender, and coagulation pathways, Wang and collaborators [33] observed that gastrointestinal cancer, infection, cisplatin therapy, vincristine use, polyurethane material, open-ended lines, and keeping time of the catheter might have an impact on PICC-VTE. Verso and colleagues [34] identified inadequate CVC tip location (seven-fold increased risk if misplaced in the upper half of SVC), left-sided CVC (five-fold increased risk), and chest radiotherapy as independent risk factors for clot formation. These data were in line with the literature [35,36,37,38]. (Figure 1).

## 4. Risk Assessment Models for PICC-Related Thrombosis in Patients with Cancer

An accurate risk assessment model (RAM) to identify cancer patients at risk for PICC-related thrombosis, who could benefit from primary thromboprophylaxis, is currently lacking (Table 1). 

One of the most validated RAM to predict VTE in ambulatory cancer patients who are starting chemotherapy, the Khorana risk score (KRS), was not specifically tested in PICC recipients [39]. The Michigan risk score (MRS), rather, considered DVT history, multi-lumen PICC, active cancer, another CVC, and white cell count > 12.0 × 10^9^/µL, and has been specifically assessed for the prediction of thrombosis in patients with PICC. In a large study on 23,010 patients (23.5% with a history of cancer and 6.2% with active cancer) whose PICC was managed in medical general wards or the intensive care unit, MRS stratified thrombotic risk into four growing VTE-risk classes: 0.9% (class 1), 1.6% (class 2), 2.7% (class 3), and 4.7% (class 4) [40]. To assess the MRS performance in combination with age-adjusted D-dimer in the prediction of upper limbs DVT (ULDVT), Kang and collaborators [41] retrospectively analyzed 2163 patients (83.2% with active cancer) with PICC. The sensitivity and specificity of both the MRS and D-dimer were 0.82 and 0.09 and 0.64 and 0.64, respectively, and the overall accuracy was low. In a retrospective, single-center study that compared 147 cancer patients receiving chemotherapy through PICC and 147 non-cancer patients receiving other therapies (chronic transfusion, long-term antibiotics) through PICC, a modified version of the MRS (mMRS ≤ 3) incorporating thrombocytosis seemed to discriminate low risk patients better than KRS [42]. A recent review which tried to compare the accuracy of the Caprini [43,44], Padua [45], Autar [46], MRS, Seeley [47], Wells [48], revised Geneva [49], and KRS scores, demonstrated that MRS was the most accurate model for the prediction of VTE in high-risk cancer patients with PICC [50]. There is limited evidence on RAM for PICC-related thrombosis in hematological malignancies. In a retrospective, single center study of 117 hematological patients with PICC, the Caprini seemed to outperform the revised Geneva, Padua, and MRS scores for the prediction of PICC-related thrombosis [51].

**Table 1 biomedicines-13-00786-t001:** Main risk assessment models to predict VTE.

Risk Assessment Model	Variables	Specific for Patients with Cancer	Specific for Patients with PICC
Khorana risk score [39,42]	Cancer site, platelet count, hemoglobin, erythropoiesis-stimulating agents, leukocyte count, BMI	Yes	No
Michigan risk score [40,41,42,50,51]	History of VTE, multi-lumen PICC, active cancer, presence of another CVC, leukocyte count	No	Yes
Caprini risk score [43,44,50,51]	Multiple variables including age, cancer, surgery, medical diseases, thrombophilia, female specific health issues, CVC	No	No
Padua risk score [45,50,51]	Active cancer, previous VTE, reduced mobility, thrombophilia, recent trauma/surgery, age, heart/respiratory failure, acute myocardial infarction/stroke, acute infection/rheumatologic disorder, obesity, hormonal treatment	No	No
Autar DVT scale [46,50]	Age, mobility, trauma, medical diseases, BMI, female specific health issues, surgery	No	No
Seeley score [47,50]	Medical diseases	No	Yes
Wells score [48,50]	Clinical symptoms for DVT, previous DVT/PE, immobility/surgery, no alternative diagnosis, heart rate > 100 beats/minute, cancer, hemoptysis	No	No
Revised Geneva score [49,50,51]	Age, previous DVT/PE, surgery/fracture, active cancer, symptoms, clinical signs	No	No

Abbreviations: VTE, venous thromboembolism; PICC, peripherally inserted central catheters; CVC, central venous catheter, BMI, body mass index; DVT, deep vein thrombosis; PE, pulmonary embolism.

## 5. Thromboprophylaxis in Cancer Patients with PICC

The role of antithrombotic prophylaxis for PICC-VTE is unclear. Various anticoagulant agents, including vitamin K antagonists (VKAs) [52], low-molecular-weight heparins (LMWHs), and direct oral anticoagulants (DOACs), have been investigated. There are no robust data yet on the possible role of factor XI inhibitors in this setting [53]. Recently, a prospective single-arm study by Pfeffer and coworkers [54] found a lower incidence in device-related VTE in 22 cancer ambulatory patients with CVC lines (one with PICC line) who underwent treatment with gruticibart, a factor XI inhibitor, as compared to the internal control study subjects (12.5% vs. 40%). The ETHIC study, which included 385 cancer patients with CVC, found similar thrombosis rates (14% and 18%, respectively, *p* = 0.35) in patients who were assigned to thromboprophylaxis with enoxaparin (40 mg/die for 6 weeks) and in those receiving placebo [55]. The efficacy and safety of primary thromboprophylaxis with DOACs in cancer patients at intermediate–high risk according to the KRS and starting systemic chemotherapy were demonstrated in the AVERT [56] and the CASSINI trials [57], even though both studies were not aimed at specifically investigating CVC-VTE. A post-hoc analysis of the AVERT trial in 217 patients with CVC (PICC lines in 72.2% of patients receiving apixaban and in 71.4% of those on placebo) showed a reduced risk of VTE with apixaban without an increase in bleeding risk [58]. This analysis included a relatively small number of patients and should be considered as a hypothesis-generating one. Also, thromboprophylaxis with rivaroxaban (10 mg/die) demonstrated a lower PICC-VTE rate as compared to enoxaparin (4000 anti-Xa IU/die) or no anticoagulation (3.4% vs. 12.4%, *p* = 0.10, *p* = 0.009, respectively), whereas both efficacy and safety appeared similar between the rivaroxaban and enoxaparin group (*p* = 0.743) [59]. The TRIM-Line randomized 105 cancer patients newly inserted with CVC (78.1% with a PICC line) to a 90-day thromboprophylaxis with rivaroxaban (10 mg/die) or placebo. VTE complications were lower in the rivaroxaban group (5.8% vs. 9.4%, HR: 0.58, 95% CI 0.14–22.5) without significant differences on bleeding complications [60]. A metanalysis including 12 RCT showed a significantly lower risk of symptomatic VTE in cancer patients with CVC receiving thromboprophylaxis with LMWH or VKA as compared to controls (RR: 0.61, 95% CI 0.41–0.88) with an absolute incidence reduction from 6.8% to 3.7% (*p* < 0.001), translating into 32 patients needed to be treated to prevent one event [61]. Li and colleagues [20] also reported a lower VTE rate in those with CVC (mostly PICC) receiving anticoagulants than those not on thrombophylaxis (7.6% vs. 13%, respectively, *p* < 0.01). Indeed, a Cochrane analysis found no VTE reduction with warfarin, and a significantly lower VTE incidence with LMWH as compared to no LMWH use (RR: 0.43, 95% CI 0.22–0.81), although no firm conclusions on safety could be drawn [62]. Overall, available evidence seems substantially in favor of primary thromboprophylaxis in cancer patients with CVC (mostly PICC) in terms of the risk–benefit ratio.

## 6. Ameliorating PICC Insertion Strategies

Several strategies have been adopted in the last two decades in order to minimize the risk of CVC-VTE [63]. A proper tip position in the SVC-right atrium junction assessed by an intracavitary electrocardiogram [64] prevents thrombosis compared to a more distal position in the lowest part of the SVC. Novel and less thrombogenic materials (silicone > polyurethane) together with proper aseptic techniques [65] and technical procedures, such as subcutaneous tunneling [66,67], could reduce thrombotic damage. Before insertion, the identification of the median nerve and the brachial artery avoids lesions of the vascular nervous bundle. Overall, proper securement, non-tapering [68], micro-introducer kits with small sample size needles, and an appropriate protection of the exit site also contribute to reduced infective complications. Notably, a safer approach worth nothing without an adequate vein selection (catheter-to-vein ratio ≤ 1/3), is an ultrasound (US)-guided venipuncture [69] in the “green zone” according to the Zone Insertion Method. In particular, a catheter caliper should be less than 50% of the vessel diameter to avert the formation of a clot [70]. 

## 7. The Novel PICC-PORT Lines

Among CVC lines, the advantages of PICCs in reducing bleeding risk and pneumothorax in comparison to PORTs [71] have been counterbalanced not only by the higher thrombotic complications [72], but also by other drawbacks. Indeed, PICCs are associated with aesthetic issues due to its partially external positioning, possible dislodgment, interference in daily activities (showering, bathing) [73], late onset wound dehiscence and pocket infections, and the requirement of weekly medications instead of monthly/bimonthly cleaning [74]. The PICC-PORT [75], a totally implanted line with the reservoir in the upper third of the upper arm, has been increasingly used in clinical practice and may potentially reduce the “social stigma” inconveniences, mainly in vulnerable female patients already with scars on the chest area (Figure 2).

A nationwide Japanese survey in 11,693 cancer patients with CVC found that, among several different implant sites (chest, neck, upper-arm, forearm), only the upper arm insertion site was associated with significantly lower complication rates compared to the chest site (7.4% vs. 5.2%, respectively, *p* = 0.010) [76]. A prospective study of 418 breast cancer patients with PICC-PORT observed overall complications in 6.9% of subjects (2.4% were ULVTE). The authors suggested that the micro-Seldinger technique with US-guidance and access performed in the Dawson’s “yellow zone” enabled an optimal catheter-to-vein ratio of <0.33 in 93% of the patients, and resulted in a low VTE rate [77]. The same authors retrospectively examined 4,480 patients with PICC-PORT (97% with cancer) and found a low rate of symptomatic VTE (2.1%) [78]. A prospective, observational study on 210 breast cancer patients showed that PICC-PORTs were better tolerated and perceived than PORTs. Acceptance of PICC-PORT was higher among women younger than 60 years, who were still working and socially active [79]. Cominacini and coworkers [80] identified a lower incidence of VTE in PICC-PORT than traditional PICC (2 and 29, respectively) with the majority of CVC-VTE being asymptomatic (Figure 3).

## 8. “Head-to-Head” Major Guidelines Comparison

Although CVC is a widely used and valuable tool in cancer care, indications from clinical practice guidelines about CVC-VTE preventive strategies are not consistent. The International Society on Thrombosis and Hemostasis advocates against routine pharmacological prophylaxis for CVC-related thrombosis in patients with cancer [81]. The American Society of Hematology does not suggest oral or parenteral prophylaxis except for high-risk patients and those with myeloma on thalidomide/pomalidomide/lenalidomide [82]. The European Society for Medical Oncology and the British Society of Hematology do not recommend routine pharmacological prophylaxis [83,84]. Overall, most guidelines do not suggest or recommend thromboprophylaxis for CVC-VTE based on uncertain evidence about the prevention of symptomatic events and concerns the cost–benefit ratio, bleeding risk, and patients’ burden. Interestingly, the same guidelines recommend primary thromboprophylaxis for ambulatory cancer patients at intermediate–high VTE risk, most of whom placed with CVC, before undergoing systemic chemotherapy. 

## 9. Optimal Suggested Management Approach for the Prevention of PICC-VTE in Cancer

Figure 4 proposed a practical three-step strategy to prevent VTE in cancer patients with PICC (Figure 4). 

First step (insertion technique optimization): insertion itself has been established as the main risk factor for the prompt formation of thrombosis during the first 24 h because of placement due to acute endothelial injury [85]. Table 1 summarizes the main aspects to consider for PICC insertion according to the following standards of care: the RaPeVA protocol (Rapid Peripheral Vein Assessment), the Dawson’s ZIM protocol (Zone Insertion Method), the ECHOTIP protocol (Protocol of ultrasound-based tip-navigation and tip-location), and the SIP protocol (Protocol Safe Insertion of PICC-PORT) [86] (Table 2). 

Second step (evaluation of indication and possible timely start of thromboprophylaxis): in patients with high VTE risk according to RAMs like KRS, thromboprophylaxis with LMWHs and DOACs, in preference to VKAs, may be started after PICC placement at the incipit of chemotherapy in the absence of active bleeding or high bleeding risk [87]. Third step (early ultrasound screening): given the higher incidence of PICC-VTE during the first 30 days following insertion, it is important to consider a close clinical monitoring with a low threshold for ultrasonography evaluation in case of signs or symptoms suspicious of thrombosis [88].

## 10. Conclusions

VTE is a common and feared complication in cancer, fostered by multiple risk factors, including CVC. In the last two decades, the use of CVC has steadily increased, reflecting its crucial role for the long-term administration of anticancer drugs and blood sampling. In particular, PICC lines afford an easily handled, cheaper, and faster insertion/removal compared with PORT, even though these advantages come at the cost of a higher thrombotic risk. Current guidelines recommend against the routine primary thromboprophylaxis for CVC-VTE in these patients. Further studies are needed to optimize risk stratification in cancer patients with CVC and evaluate the safety and efficacy of thromboprophylaxis with factor XI inhibitors. Finally, the validation of management strategies, such as the one depicted in Figure 4, may help standardizing the approach to patients with CVC and abating the burden of CVC-related thrombosis.

## Figures and Tables

**Figure 1 biomedicines-13-00786-f001:**
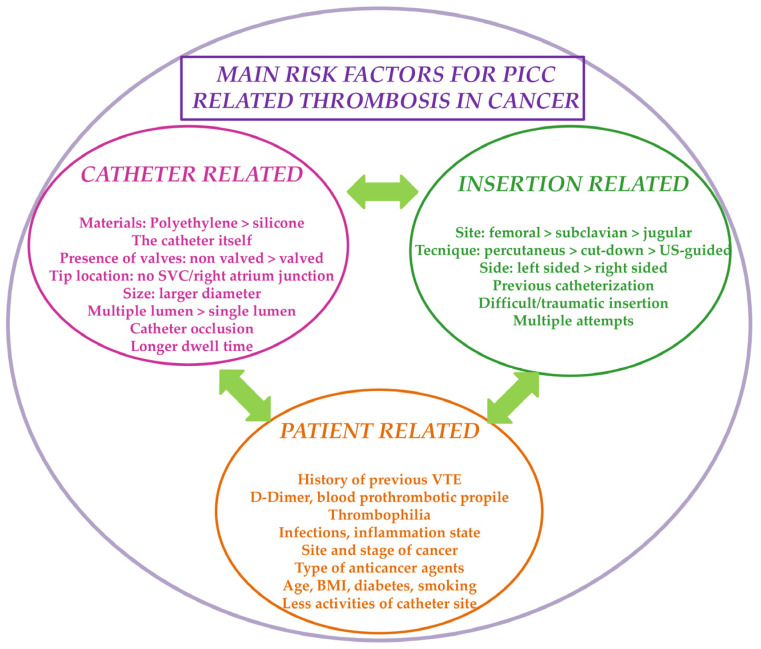
Risk factors for VTE in cancer patients indwelled with PICC. Abbreviations: VTE, venous thromboembolism; PICC, peripherally inserted central catheters; US, ultrasound; SVC, superior vena cava, BMI, body mass index.

**Figure 2 biomedicines-13-00786-f002:**
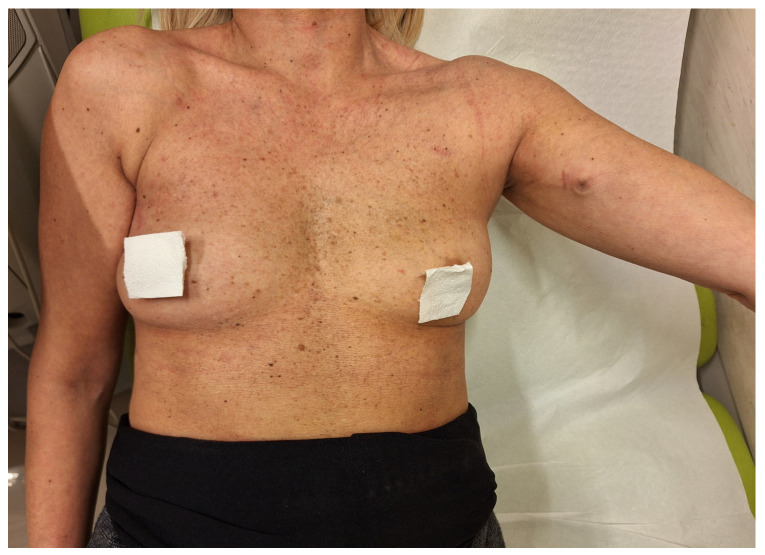
A female cancer patient placed with a left PICC-PORT line.

**Figure 3 biomedicines-13-00786-f003:**
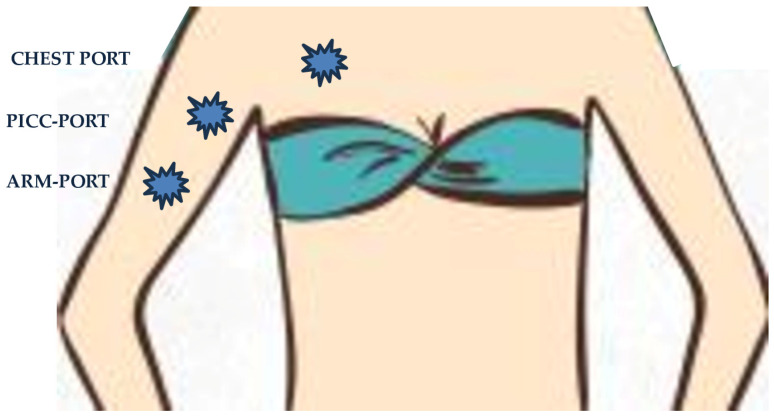
Reservoir sites for chest PORT, PICC-PORT, and arm PORT.

**Figure 4 biomedicines-13-00786-f004:**
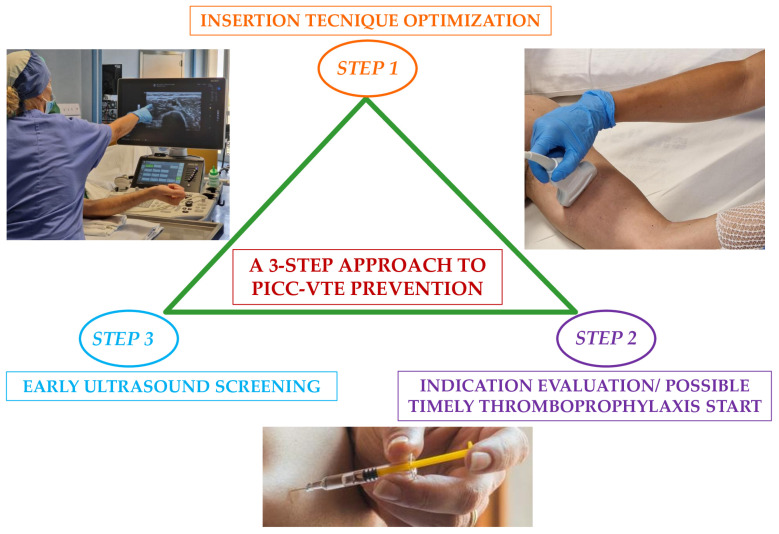
A counselled preventive three-step approach for PICC-VTE in cancer.

**Table 2 biomedicines-13-00786-t002:** Main advice for an accurate PICC insertion.

Pre-Insertion	At Insertion	Post Insertion
Ultrasound evaluation of the patency of the arm veins to rule out thrombosis	Silicone device material	Tip location at the superior vena cava/right atrium junction
Identification of the median nerve and the brachial artery	Small sample size needles	Assessment of the correct tip position by intracavitary electrocardiogram
Proper antiseptic techniques	Microintroducer kits	Proper securement
Vein (basilic, brachial) caliber selection with a catheter/vein ratio < 1/3	Ultrasound-guided venipuncture and tip navigation	Appropriate protection of the exit site
Pocket creation in the green zone (Dawson’s ZIM)	Subcutaneous tunnelling and non-tapering	Ambulatory care and maintenance of PICC line by specialist nurse team

Abbreviations: PICC, peripherally inserted central catheters; ZIM, zone insertion method.

## Data Availability

Data sharing is not applicable.

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
