# Peer review of "Prevention of Peripherally Inserted Central Catheter (PICC)-Associated Vein Thrombosis in Cancer: A Narrative Review"

_biomedicines, 2025, doi:10.3390/biomedicines13040786_

Round 1
Reviewer 1 Report
Comments and Suggestions for Authors
This is an interesting narrative review describing incidence, causes, predictive factors and possible prophylaxis of catheter related thrombosis (CRT) in cancer patients. The paper is well written, covers a wide aspect of the theory behind CRT with focus on peripherally inserted central catheters PICC and provides a comprehensive literature. Minor additions would add value and complete the paper.
Introduction: PICCs have become more popular not only in the ambulatory setting but in the stationary phase of intensive care of cancer patients. Infections are common in these immunocompromised patients and “bacterial thrombosis” can be a complication of the central vein catheters. This option should be added in the introduction in the list of causes of PICC thrombosis, because it has direct consequenses of the possible mandatory catheter removal.
Author Response
This is an interesting narrative review describing incidence, causes, predictive factors and possible prophylaxis of catheter related thrombosis (CRT) in cancer patients. The paper is well written, covers a wide aspect of the theory behind CRT with focus on peripherally inserted central catheters PICC and provides a comprehensive literature. Minor additions would add value and complete the paper.
Introduction: PICCs have become more popular not only in the ambulatory setting but in the stationary phase of intensive care of cancer patients. Infections are common in these immunocompromised patients and “bacterial thrombosis” can be a complication of the central vein catheters. This option should be added in the introduction in the list of causes of PICC thrombosis, because it has direct consequenses of the possible mandatory catheter removal.
1.I thank for the priceless attention you devoted to our paper and I totally agree with you about the need to improve the infective aspects of PICC venous thrombosis in cancer since it is the main indication, endorsed by all current guidelines, to the device removal. Accordingly, on behalf of all the coauthors, I proceed with some modifications to the manuscript in accordance with your precise indications.
2.I added 2 recent references on this topic (reference 18 and reference 19), enriching and completing the paper in the introduction section as follows.
18.Mitbander, U.B..; Geer, M.J.; Taxbro, K.; Horowitz, J.; Zhang, Q.; O’Malley, M.E.; et al. Patterns of use and outcomes of peripherally inserted central catheters in hospitalized patients with solid tumors: A multicenter study. Cancer 2022 Oct;128(20):3681-3690.
19.Sánchez Cánovas, M.; García Torralba, E.; Blaya Boluda, N.; Sánchez Saura, A.; Puche Palao, G.; Sánchez Fuentes, A.; et al. Thrombosis and infections associated with PICC in onco-hematological patients, what is their relevance? Clin Transl Oncol. 2024 Dec;26(12):3226-3235.
3.Moreover, in Figure 1 I underscored that “infections and inflammatory state” as patient’s related risk factors contribute to PICC-related thrombosis in cancer.
3.Besides, I underlined in the section on the improvements of the insertion techniques, the ones that are aimed at preventing the risk of infections beyond the thrombotic risk.
4.I added a new comparative table (Table 1, therefore the previous Table 1 becomes now Table2) on the main risk assessment models to predict venous thrombosis where acute infections is listed among the related variables
Reviewer 2 Report
Comments and Suggestions for Authors
Introduction and Significance
From the outset, the manuscript emphasizes that VTE is a major complication in oncology, particularly when central venous catheters (CVCs) are involved. PICCs hold undeniable importance for administering chemotherapy or obtaining blood samples in cancer care. However, using a PICC significantly raises the risk of thrombosis, prompting the need for an in-depth discussion of risk factors and prophylaxis options.
While the message about VTE’s high incidence and severity in cancer is accurate, it appears multiple times. A shorter, more focused introduction might better serve the reader—reserving the detailed exploration of risk factors for the dedicated sections that follow.
Risk Factors and Stratification Models
Among the most pertinent findings from the literature are the roles of obesity (body mass index over 25), a history of previous thrombosis, and improper catheter tip placement (for instance, failure to reach the cavoatrial junction). The authors also mention various other contributors, such as ovarian malignancy, repeated catheter manipulation, and left-sided insertion.
The text highlights the need for validated risk scores, such as the Khorana and Michigan Risk Scores (MRS). Although the Khorana score is widely recognized for predicting VTE in oncology, it is not necessarily tailored to PICC-related events. The Michigan Risk Score, which considers factors like previous thrombosis and multiple lumens, appears more relevant for PICCs, yet the data remain preliminary.
Prophylaxis and Ongoing Debates
The section on antithrombotic prophylaxis would benefit from a more streamlined presentation. Currently, it touches on low-molecular-weight heparins (LMWHs), vitamin K antagonists (e.g., warfarin), then switches to direct oral anticoagulants (apixaban, rivaroxaban), and finally mentions emerging factor XI inhibitors—citing studies such as ETHIC, AVERT, and TRIM-Line.
A particularly intriguing example is the TRIM-Line trial, which found that rivaroxaban (10 mg/day) reduced PICC-related thrombosis without substantially increasing bleeding risk. This suggests that prophylaxis could be considered in certain high-risk patients—for instance, those with an MRS ≥3. Still, the manuscript underscores a lack of universal consensus, given that major guidelines (ISTH, ASH, ESMO, BSH) remain cautious and generally recommend prophylaxis only in higher-risk scenarios, rather than for all individuals with a PICC.
Optimized Insertion Techniques and the PICC-PORT Concept
A relatively recent development discussed here is the PICC-PORT—a fully implanted device in the upper arm that seems to combine the cosmetic and infection-risk benefits of traditional chest ports with the straightforward insertion typical of PICCs. Preliminary data are promising, indicating fewer thrombotic events than standard PICCs, though not necessarily matching the performance of a traditional chest port.
The text also offers technical tips for correct insertion, including ultrasound-guided venipuncture, intracavitary ECG confirmation of the catheter tip, subcutaneous tunneling, and careful management of the catheter-to-vein diameter ratio. While these points are valid, they often appear as a checklist. Integrating them into a cohesive explanation of the rationale behind each measure would help clarify why these methods reduce thrombotic complications.
Guideline Comparisons and Practical Conclusions
When discussing guidelines—ISTH, ASH, ESMO, and BSH—the manuscript makes it clear that none firmly advocate routine prophylaxis for every patient with a PICC. Instead, the take-home message is that if a patient has a high thrombotic risk profile (based on clinical risk scores, tumor type, or personal history), then prophylaxis with LMWH or a DOAC is worth considering. Otherwise, widespread anticoagulation in all PICC carriers is not supported by current evidence.
The final section reiterates the need for standardized approaches and future studies on newer agents (factor XI inhibitors), ideally focusing exclusively on PICCs or on specific subgroups of cancer patients, such as those with hematologic malignancies or marked obesity.
Overall Impressions
The article is well-supported by numerous references and covers a wide spectrum of research. However, it might be improved by:
Reducing Repetition, particularly regarding the overall VTE burden in oncology patients.
Ensuring a More Coherent Structure, with clearly demarcated sections (for instance, grouping risk factors thematically or presenting prophylaxis data in a comparative table).
Providing Consistent Explanations for clinical utility (how clinicians should decide between PICCs or ports, who should receive anticoagulation, how often we should monitor for thrombosis).
Comments on the Quality of English LanguageIn medical academic writing, certain phrases often appear with slight inconsistencies or omitted prepositions. For instance:
“The increase in CVC use for long-term chemotherapies administration and blood samples is mainly related to avoidance of painful venipunctures...”
A more precise formulation would be:
“The increase in CVC use for the long-term administration of chemotherapy and for blood sampling is primarily driven by the desire to avoid painful venipunctures...”
Author Response
From the outset, the manuscript emphasizes that VTE is a major complication in oncology, particularly when central venous catheters (CVCs) are involved. PICCs hold undeniable importance for administering chemotherapy or obtaining blood samples in cancer care. However, using a PICC significantly raises the risk of thrombosis, prompting the need for an in-depth discussion of risk factors and prophylaxis options.
While the message about VTE’s high incidence and severity in cancer is accurate, it appears multiple times. A shorter, more focused introduction might better serve the reader—reserving the detailed exploration of risk factors for the dedicated sections that follow.
Risk Factors and Stratification Models
Among the most pertinent findings from the literature are the roles of obesity (body mass index over 25), a history of previous thrombosis, and improper catheter tip placement (for instance, failure to reach the cavoatrial junction). The authors also mention various other contributors, such as ovarian malignancy, repeated catheter manipulation, and left-sided insertion.
The text highlights the need for validated risk scores, such as the Khorana and Michigan Risk Scores (MRS). Although the Khorana score is widely recognized for predicting VTE in oncology, it is not necessarily tailored to PICC-related events. The Michigan Risk Score, which considers factors like previous thrombosis and multiple lumens, appears more relevant for PICCs, yet the data remain preliminary.
Prophylaxis and Ongoing Debates
The section on antithrombotic prophylaxis would benefit from a more streamlined presentation. Currently, it touches on low-molecular-weight heparins (LMWHs), vitamin K antagonists (e.g., warfarin), then switches to direct oral anticoagulants (apixaban, rivaroxaban), and finally mentions emerging factor XI inhibitors—citing studies such as ETHIC, AVERT, and TRIM-Line.
A particularly intriguing example is the TRIM-Line trial, which found that rivaroxaban (10 mg/day) reduced PICC-related thrombosis without substantially increasing bleeding risk. This suggests that prophylaxis could be considered in certain high-risk patients—for instance, those with an MRS ≥3. Still, the manuscript underscores a lack of universal consensus, given that major guidelines (ISTH, ASH, ESMO, BSH) remain cautious and generally recommend prophylaxis only in higher-risk scenarios, rather than for all individuals with a PICC.
Optimized Insertion Techniques and the PICC-PORT Concept
A relatively recent development discussed here is the PICC-PORT—a fully implanted device in the upper arm that seems to combine the cosmetic and infection-risk benefits of traditional chest ports with the straightforward insertion typical of PICCs. Preliminary data are promising, indicating fewer thrombotic events than standard PICCs, though not necessarily matching the performance of a traditional chest port.
The text also offers technical tips for correct insertion, including ultrasound-guided venipuncture, intracavitary ECG confirmation of the catheter tip, subcutaneous tunneling, and careful management of the catheter-to-vein diameter ratio. While these points are valid, they often appear as a checklist. Integrating them into a cohesive explanation of the rationale behind each measure would help clarify why these methods reduce thrombotic complications.
Guideline Comparisons and Practical Conclusions
When discussing guidelines—ISTH, ASH, ESMO, and BSH—the manuscript makes it clear that none firmly advocate routine prophylaxis for every patient with a PICC. Instead, the take-home message is that if a patient has a high thrombotic risk profile (based on clinical risk scores, tumor type, or personal history), then prophylaxis with LMWH or a DOAC is worth considering. Otherwise, widespread anticoagulation in all PICC carriers is not supported by current evidence.
The final section reiterates the need for standardized approaches and future studies on newer agents (factor XI inhibitors), ideally focusing exclusively on PICCs or on specific subgroups of cancer patients, such as those with hematologic malignancies or marked obesity.
Overall Impressions
The article is well-supported by numerous references and covers a wide spectrum of research. However, it might be improved by:
Reducing Repetition, particularly regarding the overall VTE burden in oncology patients.
Ensuring a More Coherent Structure, with clearly demarcated sections (for instance, grouping risk factors thematically or presenting prophylaxis data in a comparative table).
Providing Consistent Explanations for clinical utility (how clinicians should decide between PICCs or ports, who should receive anticoagulation, how often we should monitor for thrombosis).
Comments on the Quality of English Language
In medical academic writing, certain phrases often appear with slight inconsistencies or omitted prepositions. For instance:
“The increase in CVC use for long-term chemotherapies administration and blood samples is mainly related to avoidance of painful venipunctures...”
A more precise formulation would be:
“The increase in CVC use for the long-term administration of chemotherapy and for blood sampling is primarily driven by the desire to avoid painful venipunctures...”
1.I Thank you for the important improvements you suggested for our paper and for the detailed examination of the manuscript for which you spent precious time. On behalf of all the coauthors, I edited the manuscript in accordance with your punctual and valuable indications.
2.I reduced the length of the introduction, to offer a more incisive and focused message to the reader
- I erased the redundant parts on thrombosis burden in cancer patients in the introduction to simplify the concepts.
4.I already grouped the risk factors for PICC related thrombosis in cancer in Figure 1 into 3 main classes: related to patient, related to insertion and related to the catheter.
5.I added a new comparative table dedicated to streamline the data on the risk assessment models (Table 1, therefore Table 1 becomes now Table 2).
- Current guidelines do not endorse thromboprophylaxis for the development of venous thrombosis PICC related to all the patients as a routine indication. However, as we mentioned in reference number 20 (Li 2021) in the introduction section, the risk-benefit ratio appears favourable; indeed, this recent meta-analysis shows that cancer patients placed with central venous catheters who receive thromboprophylaxis develop less thrombotic events without increasing bleedings compared to those without. Guidelines imply that for those at high-risk for thrombosis, thromboprophylaxis could be possible but without specific indications. Our intention is to fulfill this important gap of literature with a critical reading of the published evidence to start a future debate with the readers on this topic “primary thromboprophylaxis in cancer patients placed with PICC: yes or no?”. Therefore, we specifically wrote a section dedicated to our own proposal for the opportunity of using thromboprophylaxis not as a routine strategy but only as a choice for high-risk scenarios, selected by the mentioned risk assessment models in the absence of high bleeding risk. Accordingly, we expressed our view-point with conditional terms like “proposal” or “may”. For these reasons, I edited this section adding the term “suggested” in the title and replaced Figure 4 with an edited version including the term “possible” related to the use of thromboprophylaxis. Of note, in daily practice the treating physicians use thromboprophylaxis in cancer patients with PICC in case of high thrombotic risk.
7.I edited the section on the ameliorating techniques for PICC insertion with the aim to avoid complications from a checklist approach to a more narrative and cohesive explanation on the rationale for the use of every one of them. Moreover, in Table 2 the main advices for a more secure PICC insertion are listed in 3 groups based on insertion timing: before insertion, at insertion and post insertion.
8.The purpose of our work was to focus on the best management of cancer patients with PICC to prevent vein thrombosis. Thus, we neither compare patients with PORT to those with PICC nor highlight the anticoagulant treatment of cancer patients who already have a thrombotic event, but these are very challenging issues in clinical practice as well and could we could deal with them in future reviews.
9.The monitoring of cancer patients with PICC for an early diagnosis of vein thrombosis is based on a very early use of ultrasound as we mentioned in the last section of the review in the full text, in Table 2 and in Figure 4; of note, this task is discussed in details in reference 87 and reference 89 as follows.
87.Brescia, F.; Pittiruti, M.; Spencer, T.R.; Dawson, R.B. The SIP protocol update: Eight strategies, incorporating Rapid Peripheral Vein Assessment (RaPeVA), to minimize complications associated with peripherally inserted central catheter insertion. J Vasc Access. 2024;25:5-13.
89.Debourdeau, P.; Lamblin, A.; Debourdeau, T.; Marcy, P.Y.; Vazquez, L. Venous thromboembolism associated with central venous catheters in patients with cancer: From pathophysiology to thromboprophylaxis, areas for future studies. J Thromb Haemost. 2021;19:2659-2673.
Reviewer 3 Report
Comments and Suggestions for Authors
This is an interesting narrative review evaluating the current PICC use, the PICC-related thrombosis (PICC-VTE), the associated risk factors, the risk stratification models, primary thromboprophylaxis, and the use of the novel PICC-PORT lines.
The study is well-written, complete and describes the topic perfectly with up to date references.
Author Response
This is an interesting narrative review evaluating the current PICC use, the PICC-related thrombosis (PICC-VTE), the associated risk factors, the risk stratification models, primary thromboprophylaxis, and the use of the novel PICC-PORT lines.
The study is well-written, complete and describes the topic perfectly with up to date references.
1.On behalf of all the coauthors, I thank for your technical and detailed appreciation of our paper, underscoring all the scientific issues we wanted to point out in the task of PICC related venous thrombosis in cancer, and for the precious time you devoted to our work.
Round 2
Reviewer 2 Report
Comments and Suggestions for Authors
Dear Authors,
I would like to extend my sincere thanks for addressing my comments so thoroughly. Your revised manuscript now offers a more succinct and coherent discussion of the prevention of peripherally inserted central catheter (PICC)–associated vein thrombosis in patients with cancer. Notably, the introduction is more streamlined, preventing unnecessary repetition while effectively highlighting the clinical significance of the topic.
Your reorganized sections on risk factors and prophylaxis now provide readers with a clear synthesis of the latest evidence. The addition of the comparative table on risk assessment models (RAMs) and the table outlining best practices for PICC insertion steps help clarify complex points and should facilitate practical implementation in clinical settings. The updated narrative also integrates the rationale behind each insertion technique, making the discussion both methodical and educational.
Furthermore, your balanced appraisal of prophylaxis aligns well with the most recent data and guidelines. By presenting a practical, conditional approach to anticoagulation—focusing on patients at higher risk according to RAMs—you acknowledge the nuances in current evidence and leave room for future research on emerging treatments, such as factor XI inhibitors.
Overall, the revisions have substantially strengthened both the clarity and academic rigor of the manuscript. In my opinion, it now meets the standards expected for publication. Congratulations on this achievement, and I look forward to seeing this work disseminated to the broader medical community.